# Antidepressive-like Behavior-Related Metabolomic Signatures of Sigma-1 Receptor Knockout Mice

**DOI:** 10.3390/biomedicines10071572

**Published:** 2022-07-01

**Authors:** Baiba Svalbe, Baiba Zvejniece, Gundega Stelfa, Karlis Vilks, Edijs Vavers, José Miguel Vela, Maija Dambrova, Liga Zvejniece

**Affiliations:** 1Laboratory of Pharmaceutical Pharmacology, Latvian Institute of Organic Synthesis, Aizkraukles Street 21, LV-1006 Riga, Latvia; baiba.zvejniece@farm.osi.lv (B.Z.); gundega@farm.osi.lv (G.S.); karlis.vilks@farm.osi.lv (K.V.); edijs.vavers@farm.osi.lv (E.V.); maija.dambrova@farm.osi.lv (M.D.); liga@farm.osi.lv (L.Z.); 2Faculty of Medicine, University of Latvia, Raiņa Boulevard 19, LV-1586 Riga, Latvia; 3Faculty of Veterinary Medicine, Latvia University of Life Sciences and Technologies, K.Helmana Street 8, LV-3004 Jelgava, Latvia; 4Institute of Solid State Physics, University of Latvia, 8 Kengaraga Street, LV-1063 Riga, Latvia; 5Faculty of Pharmacy, Riga Stradins University, Dzirciema Street 16, LV-1007 Riga, Latvia; 6Welab Barcelona, Barcelona Science Park (PCB), Baldiri Reixac 4-8, 08028 Barcelona, Spain; jvela@welab.barcelona

**Keywords:** metabolomic signatures, sigma-1 receptor, despair behavior, ceramide, serotonin

## Abstract

Sigma-1 receptor (Sig1R) has been proposed as a therapeutic target for neurological, neurodegenerative, and psychiatric disorders, including depression and anxiety. Identifying metabolites that are affected by Sig1R absence and cross-referencing them with specific mood-related behaviors would be helpful for the development of new therapies for Sig1R-associated disorders. Here, we examined metabolic profiles in the blood and brains of male CD-1 background Sig1R knockout (KO) mice in adulthood and old age and correlated them with the assessment of depression- and anxiety-related behaviors. The most pronounced changes in the metabolic profile were observed in the plasma of adult Sig1R KO mice. In adult mice, the absence of Sig1R significantly influenced the amino acid, sphingolipid (sphingomyelin and ceramide (18:1)), and serotonin metabolic pathways. There were higher serotonin levels in plasma and brain tissue and higher histamine levels in the plasma of Sig1R KO mice than in their age-matched wild-type counterparts. This increase correlated with the reduced behavioral despair in the tail suspension test and lack of anhedonia in the sucrose preference test. Overall, these results suggest that Sig1R regulates behavior by altering serotonergic and histaminergic systems and the sphingolipid metabolic pathway.

## 1. Introduction

Sigma-1 receptors (Sig1R) are transmembrane chaperone proteins localized in mitochondria-associated endoplasmic reticulum membranes (MAMs) and expressed in the central nervous system and peripheral organs such as the liver, spleen, adrenal gland, lung, and heart [1,2,3]. Sig1R is considered a pluripotent regulator of various cellular functions, including calcium transport, stress response, neuroplasticity, and lipid metabolism [4,5]. Based on experimental animal studies, Sig1R is known to play a role in neurodegenerative diseases such as amyotrophic lateral sclerosis, Alzheimer’s, and Huntington’s disease, ischemia–reperfusion damage, pain, drug addiction, and psychiatric disorders [6,7,8,9]. Recently, Sig1R has been suggested as a potential drug target against inflammatory conditions and cancer [10,11] in addition to cardiovascular diseases and COVID-19 [10,12].

Since the interaction between Sig1R and numerous antidepressant drugs and neurosteroids was discovered, much evidence has linked Sig1R and mood disorders [13]. Genome-wide association studies in the general population of humans have not found a link between SIGMAR1 and major depressive disorder [14,15]. However, a SIGMAR1 polymorphism (rs1800866) that results in a substitution from glutamine (CAG) to proline (CCG) has been associated with an increased risk of major depressive disorder in the Japanese population [16] and bipolar disorder in the Korean population [17].

To better understand the physiological role played by Sig1R, Langa et al. generated Sig1R knockout (KO) mice by homologous recombination [18]. Sig1R KO mice were developed using albino outbred CD-1 and inbred C57BL/6J background mice [18]. Only female C57BL/6J Sig1R KO mice showed decreased spatial and working memory and altered stress-conditioned memory [19]. Depression- and anxiety-like behavior was observed in male C57BL/6J Sig1R KO mice in the forced swimming, tail suspension, and elevated plus maze tests [19,20,21,22,23]. It should be noted that depression-like behaviors in genetically modified mice could depend on inbred and outbred background strains [24]. There are a limited number of long-term phenotyping studies performed with CD-1 background Sig1R KO animals, and depression-related behavior has been poorly evaluated in these mice [9]. Recently, the first long-term study demonstrated that reduced behavioral despair could be a specific characteristic of CD-1 background Sig1R KO mice [25]. However, this was observed in Sig1R KO mice following TBI [25]. It is therefore essential to evaluate depression- and anxiety-related behavior at different ages in the same animals because mood-related disorders can develop over a lifetime [26]. Performing long-term phenotyping in CD-1 background KO animals would thus be critical to fully understanding the involvement of Sig1R in physiopathology.

Metabolomics is a powerful tool used to assess altered signaling pathways and to determine affected metabolic pathways in various tissue samples at different ages in transgenic animals [27]. Accordingly, it may provide a biochemical/neurochemical correlate to findings in behavioral studies. To date, comprehensive metabolomic profiling has not been carried out in Sig1R KO mice. Only a few changes in liver metabolites and metabolites associated with cellular oxidative stress have been reported in Sig1R KO animals [28].

The precise mechanisms through which Sig1R modulates depressive behavior have not yet been elucidated. The present study aimed to examine metabolic profiles in the blood and brain of male CD-1 background Sig1R KO mice in adulthood and old age and correlate them with the assessment of depression- and anxiety-related behaviors. In addition, the voluntary motor activity and the eating and drinking behavior of Sig1R KO and WT mice were evaluated, and the expression of selected genes and proteins was investigated by quantitative PCR and immunohistochemistry. We found metabolomic signatures associated with reduced behavioral despair in plasma and brain tissue in Sig1R KO mice.

## 2. Materials and Methods

### 2.1. Animals

All studies involving animals are reported in accordance with ARRIVE guidelines [29]. Experimental procedures were performed in accordance with the guidelines reported in the EU Directive 2010/63/EU and local laws and policies. All procedures were approved by the Latvian Animal Protection Ethical Committee of Food and Veterinary Service in Riga, Latvia. Fifty-three age-matched CD-1 background male wildtype (WT) (ENVIGO, Venray, Netherlands) and 50 Sig1R knockout (Sig1R KO) mice (ENVIGO, Bresso, Italy), generously provided by Esteve Pharmaceuticals, were used in this study. The description of the phenotype of the Sig1R KO mice has been previously described [18,30]. The mice were 6–7 weeks old at the start of the experiment; all animals were housed under standard conditions (21–23 °C, 12 h dark-light cycle) with unlimited access to standard food (Lactamin AB, Mjölby, Sweden) and water in an individually ventilated cage housing system (cage size 38 cm × 19 cm × 13 cm, Allentown Inc., Allentown, NJ, USA). Each cage contained bedding consisting of EcoPure™ Shaving’s wood chips (Datesand, Cheshire, UK), nesting material, and a wooden block from TAPVEI (TAPVEI, Paekna, Estonia). For enrichment, transparent tinted (red) nontoxic durable polycarbonate Safe Harbor Mouse Retreat (Animalab, Poznan, Poland) was used. Mice were housed with three to five animals per standard cage. The WT and Sig1R KO mice were housed separately. During the study, three animals died by natural death and were discovered during daily monitoring. Two WT mice died at the 63rd and 75th week of age, respectively, and one Sig1R KO mouse died at the 30th week of age.

### 2.2. Experimental design

Behavioral testing was performed by experienced testers blinded to the experimental group. Behavioral performance and physiology tests were carried out in 2- to 18-month-old male mice during the dark phase of the dark–light cycle (lights off at 08:00 h and on at 20:00 h). The mice were weighed once a week in the afternoon throughout the study. Following behavioral analysis, mice at 4 months of age were sacrificed to perform real-time quantitative PCR (RT–qPCR) analysis, and mice at 6 and 18 months of age were sacrificed to perform immunohistochemical analysis and quantify different metabolites in blood and brain samples. The experimental schedule is shown in Figure 1.

### 2.3. Detection of Endogenous Metabolites

Blood plasma and brain tissue were collected from 6- and 18-month-old animals and profiled using a commercially available MxP^®^ Quant 500 kit, which detects 630 metabolites from 26 biochemical classes (BIOCRATES Life Sciences AG, Innsbruck, Austria). Mice were anaesthetized using i.p. administration of ketamine (200 mg/kg) and xylazine (15 mg/kg). The depth of anesthesia was monitored by toe pinch using tweezers. The blood was collected from the left ventricle before the perfusion. Animals were transcardially perfused with 15 mL of 0.01 M phosphate buffered saline (PBS, pH = 7.4) to remove the remaining blood from the tissue. Frozen samples (50–70 µL) were sent to BIOCRATES Life Sciences AG. Data of all measured metabolites were used as received; no data correction was applied except for the exclusion of metabolite data under the level of detection. A specific list of the assays performed and detected metabolites in plasma and brain tissue are documented in Appendix A, and further details regarding the methods used to conduct each assay are available at https://biocrates.com/ (accessed on 26 June 2019).

### 2.4. Voluntary Activity and Ingestion Behavior

The voluntary activity in the running wheel and the feeding and drinking behavior of Sig1R KO and WT mice were evaluated at 2, 4, 8, 10, and 14 months of age using the metabolic cages of the PhenoMaster system (TSE Systems, Bad Homburg, Germany) [31]. The behavior of animals was continuously recorded for 72 h with the following parameters taken every 15 min: the voluntary running wheel (rotations per minute (rpm)), daily food consumption (g), and daily water consumption (mL), with these two final parameters being adjusted for body weight and energy expenditure (kcal/h/g). The voluntary running wheel activity was evaluated at 2, 4, and 14 months of age.

### 2.5. Depression-like Behavior

#### 2.5.1. Tail Suspension Test

The tail suspension test was performed at 2, 6, and 15 months of age to detect a depressive state in mice as described previously [25,32]. To prevent tail climbing behavior, a plastic tube (4 cm long) was put on each mouse’s tail. Each animal was suspended with tape (17 cm) from a horizontal rod elevated 50 cm above the table. To prevent mice from interacting with each other, the box was divided by a wall. Mice were recorded for 6 min using the digital HD video camera recorder (Handycam HDR-CX11E, Sony Corporation, Tokyo, Japan) and immobilization was analyzed during the final 4 min.

#### 2.5.2. Sucrose Preference Test

The preference of the mice for sucrose-sweetened versus tap water was measured at 2 and 14 months of age as described by Koehl [33] with slight modifications. Mice were individually housed and acclimated to PhenoMaster system cages with food and water for two days. For the baseline measurements, mice had free access to two drinking bottles containing tap water for 48 h. Animals were then exposed to two drinking bottles, one with tap water and the other with 1.6% fresh sucrose solution, for 48 h. The preference of the mice for 1.6% sucrose or tap water was measured from 24 to 48 h. The position of the sucrose and tap water bottles were switched every 12 h to avoid place preference. The total consumption of water and sucrose solution was taken every 24 h and was used for analysis. Sucrose preference was calculated using the following formula: (sucrose solution intake (mL)/total fluid intake (mL)) × 100.

### 2.6. Anxiety-like Behavior 

#### 2.6.1. Zero Maze Test

To evaluate anxiety, the Zero maze test was performed in 2-, 6-, and 17-months old mice. The Zero maze was constructed of a black circular track 4,5 cm wide, 50 cm in diameter, and elevated 44 cm above the floor. The experiment was conducted under dim lighting (46 Lux). The animals were gently placed in the center of an open area, and time spent in the open and close arm was registered for 5 min using an EthoVision video tracking system (version XT 11,5; Noldus, Wageningen, The Netherlands). Between trials, the maze was cleaned with 70% ethanol.

#### 2.6.2. Open-Field Test

To detect the locomotor activity of the mice at 2, 10, and 13 months of age, the open-field test was performed as described previously [34]. In addition, time spent by mice in the center zone was recorded as a measure of anxiety-like behavior. The test apparatus was a square arena (44 cm × 44 cm) with a black floor. The experiment was conducted under dim lighting. The animals were gently placed in the center of the field, and their behavioral parameters were recorded using an EthoVision video tracking system (version XT 11,5; Noldus, Wageningen, The Netherlands). The moved distance (cm), velocity (cm/s), vertical activity (number), and time spent in the center (s) were recorded. The testing consisted of a 16-min session for each mouse. 

### 2.7. Quantitative PCR (qPCR) Analysis

The total RNA from the brain substructures was isolated using a PureLink™ RNA Mini Kit (Thermo Fisher Scientific, Waltham, MA, USA), and first-strand cDNA synthesis was carried out using a High Capacity cDNA Reverse Transcription Kit (Applied Biosystems™, Foster City, CA, USA) following the manufacturer’s instructions. qPCR analysis of gene expression was performed by using an SYBR^®^ Green Master Mix (Applied Biosystems™, Foster City, CA, USA) or QuantiNova SYBR^®^ Green PCR (Qiagen, Hilden, Germany) mix with synthesized cDNA and forward and reverse primers. Primers were designed using the Primer-BLAST tool [35] and are listed in Appendix A. Reactions were run on a Bio-Molecular Systems MIC qPCR Cycler according to the manufacturer’s protocol. The relative expression levels for each gene were calculated with the ΔΔ*C_t_* method and normalized to the expression of the housekeeping gene *β-actin*. 

### 2.8. Weight of Organs

At the age of 4, 6, and 18 months, the mice were sacrificed by decapitation. The brain, kidney, liver, heart, and pancreas weights of WT and Sig1R KO animals were measured (KERN & SOHN GmbH, Balingen, Germany), and organ-to-body weight ratios were calculated.

### 2.9. HistologyBrain Tissue Preparation for Histological Analysis

#### Brain Tissue Preparation for Histological Analysis

Mice were anaesthetized using i.p. administration of ketamine (200 mg/kg) and xylazine (15 mg/kg). The depth of anesthesia was monitored by toe pinch using tweezers. Animals were transcardially perfused with 15 mL of 0.01 M phosphate-buffered saline (PBS, pH = 7.4) to remove the blood from the tissue. After perfusion, brains were carefully dissected and post-fixed in 4% PFA overnight at 4 °C. Brains were cryoprotected with a 10–20–30% sucrose gradient in 0.01 M PBS over 72 h. Samples were embedded in an optimal cutting temperature compound (Tissue-Tek^®^ O.C.T.™ Compound, Sakura Finetek Europe B.V., Alphen aan den Rijn, Netherlands) and placed in a dry ice/isopropanol slurry. Frozen samples were stored at −80 °C. Coronal sections (35 μm thick) of brain were cut using a cryostat Leica CM1850 (Leica Biosystems, Buffalo Grove, IL, USA) and stored in antifreeze buffer at −20 °C until staining was performed.

### 2.10. Immunohistochemistry

#### 2.10.1. Staining of Serotonin (5-HT) and Tryptophan Hydroxylase 2 (TPH2)

The staining of free-floating sections was performed based on a method described previously [36]. The following primary antibodies were used in this study: 5-HT (Serotonin) rabbit antibody (1:10,000, +4 °C, overnight; ImmunoStar, Hudson, WI, USA, Cat# 20080) and recombinant anti-TPH2 antibody [EPR19191] (1:1000, room temperature, overnight; Abcam, Cambridge, UK, Cat# ab184505). Biotinylated goat anti-rabbit IgG (H + L) antibody (1:1000, Invitrogen, Waltham, MA, USA, Cat# 65-6140) was used as a secondary antibody. Streptavidin conjugated with horseradish peroxidase (1:1000, Abcam, Cat# ab7403) and 3,3′-diaminobenzidine tetrahydrochloride (DAB, Sigma-Aldrich, Cat# D5905) and hydrogen peroxide was used to reveal peroxidase immunostaining. DAB-stained sections were mounted on gelatinized slides, cleared in xylene, mounted using DPX mounting medium (Sigma-Aldrich, St. Louis, MO, USA), coverslipped, and finally photographed and measured. Images were taken with a Nikon Eclipse TE300 microscope (Nikon Instruments, Tokyo, Japan).

#### 2.10.2. Image Analysis 

Brain sections corresponding to identical anatomical structures were used for the analysis. The structures were validated using the Allen Mouse Brain atlas (http://mouse.brain-map.org/static/atlas, accessed on 18 May, 2020). For each staining experiment, sections from all animals were processed in the same staining tray. In the quantification experiments, each hemisphere was analyzed separately. Staining was quantified using ImageJ software (ImageJ v1.52a). Eight-bit images were generated from the pictures and were cropped to contain the regions of interest. Means of optical density (OD) were used to quantify the staining intensity of 5-HT in specific brain structures. For OD analysis, calibration was performed in accordance with the instructions on the ImageJ software website (https://imagej.nih.gov/ij/docs/examples/calibration/, accessed on 10 August 2020). Negative controls replacing the primary antibody with a buffer solution only (TBS-T) were performed. Obtained OD values of corresponding background staining were subtracted from the total OD values measured for each brain structure. Images for TPH2 staining were thresholded to select a specific signal over the background, and the number of stained TPH2-positive neurons for each region was counted using ImageJ software.

### 2.11. Statistical Analysis

Statistical analysis was performed using GraphPad Prism (GraphPad Software, Inc., La Jolla, CA, USA). Two-way analysis of variance (ANOVA) or mixed effects models were used to detect multiple factors. Sidak’s or Fisher’s multiple comparisons tests were used. 5-HT and TPH2 staining was analyzed using ordinary two-way ANOVA followed by Tukey’s test. Differences between two groups were assessed using the Student’s *t*-test. *p*-values less than 0.05 were considered significant. Data are shown as mean ± SEM.

## 3. Results

### 3.1. Age- and Genotype-Related Metabolic Signatures in Plasma and Brain Tissue of Sig1R KO Mice

We analyzed the concentrations of metabolites in the plasma and brain tissues of mice in adulthood and old age (Figure 2). In the final dataset, of the 630 metabolites, 373 were identified in the plasma and 179 in the brain tissue (Appendix A). Hierarchical clustering analysis of plasma samples revealed clustering between adult Sig1R KO and WT mice but not between the old animals. In addition, the results from the brain samples of the hierarchical clustering analysis illustrated a group separation between the Sig1R KO and WT mice, except for one sample at each age.

Multivariate analyses of metabolomics data, including heatmap, volcano plot, and principal components analysis, revealed a significant difference in metabolite concentrations in plasma and brain tissues between Sig1R KO and WT mice (Figure 2). Adult Sig1R KO mice showed a significant decrease of 70 metabolites and an increase of 12 metabolites in the plasma samples (Figure 2A, Appendix A). Plasma from adult Sig1R KO mice had significantly decreased concentrations of amino acids (AAs), both essential and nonessential, as well as short-chain acylcarnitines, lysophosphatidylcholines, phosphatidylcholines, sphingomyelins, ceramides, and hexosylceramides (Figure 2A). Moreover, in the plasma of adult Sig1R KO mice, there were significantly higher levels of histamine, anserine, and carnosine and lower levels of dimethylarginine (ADMA), phosphatidylcholine (32:1 (PCaa)), taurine, ceramide (d18:1 (Cer)), and sphingomyelin (C16:1 (SM)) than in the plasma of adult WT mice. In contrast, in humans, these metabolites were shown to decrease in individuals with depression [37]. The changes in the differences in metabolites were less pronounced in the plasma of old Sig1R KO mice than in that of WT mice (Figure 2B and Appendix A). The concentrations of histamine and serotonin were significantly higher in the plasma of old Sig1R KO mice than in that of WT mice (Figure 2B). The results of the volcano plot analysis showed that 76 metabolites were significantly differentially expressed between adult Sig1R KO and WT mice (FC > 1.2; *p* < 0.05), including 11 upregulated and 56 downregulated metabolites (Figure 2C). Similar results arising from the volcano plot analysis are shown in old mouse plasma (Figure 2D).

In addition, we analyzed the concentrations of metabolites in the brain tissues of Sig1R KO and WT mice. The brains of the adult Sig1R KO mice were characterized by significantly lower levels of some sphingomyelins (SM), arachidonic acid (AA), and amino acids and higher levels of several hexosylceramides (HexCer), ceramides, and phosphatidylcholines (PC) (Figure 2G, Appendix A). The brains of old Sig1R KO mice were characterized by significantly lower concentrations of several long-chain acylcarnitines and choline and significantly higher concentrations of several ceramides, carnitine, short-chain acylcarnitines, amino acids, and aconitic acid (Figure 2H, Appendix A). The results of the volcano plot analysis of the brain tissues showed that 24 metabolites were differentially expressed between adult Sig1R KO and WT mice (FC > 1.2; *p* < 0.05), including 21 upregulated and 3 downregulated metabolites (Figure 2K). Brain tissues from the old mice showed similar results in the volcano plot analysis (Figure 2L).

Principal component analysis (PCA) showed that Sig1R KO mice have more affected metabolites in blood plasma than in the brain both during adulthood and old age (Figure 2E,F,K,L). The metabolic pathway analysis identified 35 altered metabolic pathways based on the metabolite concentrations in the plasma of adult Sig1R KO mice. The top-ranked altered metabolic pathways were as follows: phenylalanine metabolism; aminoacyl-tRNA biosynthesis; histidine metabolism; phenylalanine, tyrosine, and tryptophan biosynthesis; alanine, aspartate, and glutamate metabolism; glycine, serine, and threonine metabolism; and taurine and hypotaurine metabolism (Figure 2M, Appendix A). Quantitative enrichment analysis indicated that these pathways of metabolism dysregulation are associated with various kinds of seizures and neurological, cardiovascular, and lipid diseases (Figure 2N). These findings suggest that Sig1R is involved not only in central nervous system processes but also in cardiovascular function and lipid metabolism.

### 3.2. Reduced Behavioral Despair in Sig1R KO Mice Is Associated with Increased Levels of Serotonin

Comprehensive behavioral phenotyping showed that the most pronounced changes occurred in depression-related behavior in young, adult, and old Sig1R KO mice compared to WT mice. In the tail suspension test, Sig1R KO mice showed reduced behavioral despair, illustrated by a significantly decreased immobility time compared to WT mice in all age groups (Figure 3A). Using the sucrose preference test, anhedonia was observed only in old WT but not old Sig1R KO mice (Figure 3C–E). The preference for the sucrose solution was significantly higher in old Sig1R KO mice than in WT mice (Figure 3F).

Metabolomics data showed that in adult animals, there was an observed tendency of higher serotonin concentration in blood plasma, while the concentration of serotonin in plasma was significantly increased in old Sig1R KO compared to old WT mice (Figure 3G). In addition, the serotonin concentration was significantly higher in the brain cortex of adult Sig1R KO mice than in that of adult WT mice (Figure 3H). Furthermore, depression-like behavior has been associated with decreased serotonin concentrations in plasma and brain tissue [38].

To compare the differences in the expression of the serotoninergic neuron marker TPH2 and the levels of 5-HT in different brain regions between WT and Sig1R KO animals, we performed immunohistochemical staining in adult and old WT and Sig1R KO mice. We analyzed different brain structures that receive serotoninergic input from the dorsal and medial raphe nuclei. Both adult and old Sig1R KO mice demonstrated significantly increased immunoreactivity for 5-HT-containing neurons in the cortex (+0.8 mm from bregma) by 1.3- and 1.8-fold, respectively (Figure 2L). No significant differences in immunoreactivity for 5-HT were found in the hippocampus, amygdala, hypothalamus, striatum, or dorsal raphe nucleus between WT and Sig1R KO animals (data not shown). TPH2 staining showed that the number of serotoninergic neurons was not different in the dorsal raphe nucleus between WT and Sig1R KO animals (neither adult nor old, Figure 3M,N, Appendix A), indicating that the difference in the level of 5-HT in the brain might occur due to differences in the serotonin release/uptake mechanism.

Next, we explored 5-HT receptor gene expression in different brain structures in adult mice. We measured the expression of *Htr1a*, *Htr1b*, *Htr2a*, *Htr2c*, and *Htr3a* in the prefrontal cortex, striatum, midbrain, hypothalamus, hippocampus, and cortex (Figure 3B). In the prefrontal cortex, Sig1R KO mice had significantly higher expression of the gene *Htr3a*, which modulates depression- and anxiety-related behaviors [39]. The gene expression of *Htr1a*, *Htr1b*, *Htr2a*, and *Htr2c* was not significantly altered. Similar to the immunohistochemistry results, the mRNA level of *Tph2*, which is the rate-limiting enzyme in the synthesis of serotonin, was not affected in different brain structures in adult Sig1R KO mice (Table 1). In the cortex, Sig1R KO mice had significantly higher expression of the gene *Gabbr2* (Table 1). The gene expression of dopamine receptors, which could be involved in depression-related behavior, did not change in Sig1R KO mouse brain structures compared to WT mouse brain structures (Table 1).

In summary, Sig1R KO mice showed reduced behavioral despair and a lack of anhedonia compared to WT mice, which may be associated with increased plasma and brain tissue serotonin levels.

### 3.3. Anxiety-like Behavior Characteristics of CD-1 Background Sig1R KO Mice

We evaluated anxiety-like and locomotor behavior in young, adult, and old Sig1R KO and WT mice in zero-maze and open-field tests and voluntary activity in the metabolic cages by using running wheels.

Both adult and old Sig1R KO mice displayed anxiogenic-like behavior compared to age-matched Sig1R KO mice, as revealed by a significant decrease in time spent in the center of the open-field (Figure 4A,B), with no apparent difference in locomotor activity (Figure 4C). Voluntary activity was evaluated in young, adult, and old mice in the metabolic cages with a freely accessible running wheel for a total of 72 h. Immediately after being placed in the metabolic cages, during the first six hours, young and adult Sig1R KO mice were significantly more active than age-matched WT mice (Figure 4D,E). However, the total 72 h activity did not differ between Sig1R KO mice and age-matched WT mice (data not shown). These results indicate that Sig1R KO mice demonstrate increased first 6 h motor activity, which could be associated with anxiogenic-like behavior. There were no differences in the time spent in the open area of the zero-maze when comparing age-matched Sig1R KO and WT mice (Figure 4F).

### 3.4. Physiological and Anatomical Characteristics of Sig1R KO Mice

During the study, we monitored the physiological and anatomical parameters of all animals. Sig1R KO mice gained weight more slowly than WT mice only in the first 4 months (Figure 5A). However, the amount of food consumed did not differ between Sig1R KO mice and age-matched WT mice (data not shown). Moreover, from 5 months of age and throughout their lifetime, we did not observe significant body weight differences between Sig1R KO and WT mice. The energy expenditure was also not altered by Sig1R gene knockout in young (Figure 5B) or old age (data not shown). Sig1R KO mice had heavier brains than WT mice in adulthood and old age; however, the weights of the heart, kidneys, and pancreas were significantly lower (Table 2).

## 4. Discussion

To date, metabolomic analysis has been widely used to identify metabolites involved in psychiatric diseases such as depression and anxiety [37]. This study showed that male Sig1R KO mice with a CD-1 background exhibit reduced behavioral despair throughout their lifespan. In addition, WT mice, but not Sig1R KO mice, developed anhedonia with age. We also observed that Sig1R KO mice showed anxiogenic-like behavior. By performing simultaneous metabolomic and long-term phenotyping studies in CD-1 background Sig1R knockout animals, we revealed a significant link between several metabolomic signatures and the specific phenotypes of these animals.

Ceramides are new biomarkers for depression [40,41,42], mainly based on the observation that ceramide concentrations are increased in the blood plasma of patients with depression [42,43]. In parallel, it has been suggested that Sig1R is involved in regulating lipid synthesis [44,45,46]. Accordingly, Sig1R knockdown suppressed glucosylceramide synthesis in Chinese hamster ovary cells [46], thereby regulating cellular levels of ceramide. In our study, ceramide concentrations were decreased by approximately 25% in adult Sig1R KO mouse plasma. Serine, an important amino acid for de novo ceramide biosynthesis in the endoplasmic reticulum [47], was also decreased in adult Sig1R KO mouse plasma (Figure 2A). Therefore, it is possible that reduced behavioral despair in Sig1R KO mice could be associated with decreased ceramide concentrations in plasma. It is also known that mice overexpressing acid sphingomyelinase displayed greater ceramide (C16:0 and C18:0) production in the hippocampus, which induced depression-like behavior [48]. Furthermore, it was shown that C16 ceramide induced a depressive- and anxiogenic-like phenotype when infused into the dorsal hippocampus or the basolateral amygdala, respectively [49]. In our study, Sig1R KO mice had an increased concentration of ceramide (18:1) in brain cortex tissue, but no depression-like behavior was observed. Similarly, after C8 and C20 ceramide infusion in the male C57BL/6J mouse brain, a depressive-like phenotype was not observed [49]. This suggests that specific ceramide species are involved in the development of depression. Overall, altered concentrations of sphingolipids (sphingomyelin and ceramide (18:1)) could be related to the reduced behavioral despair observed in Sig1R KO animals.

The metabolomic study showed that the serotonin concentration was increased in the plasma and brains of Sig1R KO mice, which was associated with reduced behavioral despair. This is in agreement with our previous study, in which CD-1 Sig1R KO mice did not develop depression-like behavior after traumatic brain injury during the 1-year follow-up monitoring [25]. Previous studies reported that C57BL/6J background Sig1R KO male mice exhibited a depressive-like phenotype [19,20,21,23]. It was also reported that SIGMAR1 plays a role in depression-related disorders in a relatively small, geographically restricted human population [16,17], but it was not associated with major depressive disorder in genome-wide association studies in the general population [15]. Similarly, depression-related behavior in Sig1R KO mice is affected by background; behavioral despair is observed in C57BL/6J but not in CD-1 background mice. The genetic variability in outbred CD-1 strain mice is higher and more closely mimics that of human populations [50,51]. The observed variations in depression-related behavior between CD-1 and C57BL/6J mice could be due to differences in the serotonergic system. A previous study showed that C57BL/6J mice express the high-functioning Tph2 allele that influences serotonin transporter function in brain tissue compared to inbred DBA/2J mice [52]. However, there is insufficient information regarding 5-HT-linked gene polymorphisms in CD-1 mice to attribute differences to larger genetic diversity in outbred mice.

The lateral habenula is a key regulator of the activity of dorsal and median raphe nuclei in which most serotoninergic neurons originate [53]. Previously, it was observed that CD-1 background Sig1R KO animals demonstrate decreased expression of the R2 subunit of GABA-B receptors in the habenula [36]. It is known that a hyperactive lateral habenula is involved in the etiology of depressive-like behaviors through the inhibition of the release of serotonin [54,55]. In our study, we found increased serotonin levels in the cortex of Sig1R KO animals, which might indicate that the overall activity of the lateral habenula could be decreased in Sig1R KO animals. Therefore, increased basal serotonin levels in the cortex of the Sig1R KO mice forebrain could explain the reduced behavioral despair observed in these animals.

In our study, metabolomic analysis revealed a 2-fold increase in the plasma histamine concentration in adult and old Sig1R KO mice. Histamine plays an important role in anxiety and depression [56]. In addition, reduced histamine receptor function yields a depression-like phenotype [57], and a decreased concentration of histamine contributes to increased anxiety-like behavior [57,58,59]. In addition, we found that 16 amino acids were significantly decreased in the plasma of adult Sig1R KO mice. Our pathway analysis revealed that aminoacyl-tRNA biosynthesis was one of the most affected pathways, which previously reflected a disturbance in amino acid metabolism [60]. In addition, the aminoacyl-tRNA biosynthesis pathway has been associated with major depressive disorder [61]. Moreover, amino acids are important precursors of the synthesis of nitrogenous compounds and hormones, which have very important biological functions in the growth, development, reproduction, and homeostasis of organisms [60]. Altogether, we think that the aminoacyl-tRNA biosynthesis pathway and increased histamine concentration could also be associated with the reduced behavioral despair observed in Sig1R KO mice.

In our study, there was no clear evidence of increased anxiety in Sig1R KO mice which were evaluated in the zero-maze test. However, Sig1R KO mice exhibited a significantly decreased amount of time spent in the center of the open-field and increased locomotor activity in the running wheel in the first 6 out of 72 h. Previously, it was shown that s1r + 25/+25 larvae (Sig1R KO zebrafish) had increased global locomotor activity following sudden placement in a new environment and changes in light intensity, but the hyperlocomotor response was not related to visual deficit or spontaneous hyperactivity [62]. The increased locomotor activity observed in both Sig1R knockout zebrafish and mice strengthens the notion about the involvement of Sig1R in the regulation of locomotion in response to a new stimulus and a new environment. The increased locomotor activity may not be directly related to anxiety but rather to another specific phenotype that requires further evaluation.

Overall, our findings suggest that the reduced behavioral despair in CD-1 background Sig1R KO male mice is related to the serotonergic and histaminergic systems and the sphingolipid metabolic pathway. We showed, for the first time, that Sig1R is involved in lipid synthesis through sphingolipid pathways in vivo in Sig1R KO mice.

## Figures and Tables

**Figure 1 biomedicines-10-01572-f001:**
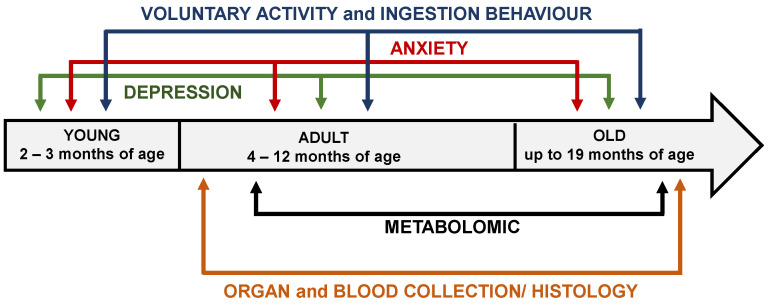
Experimental schedule of behavioral tests and metabolomic study performed in Sig1R KO and WT mice.

**Figure 2 biomedicines-10-01572-f002:**
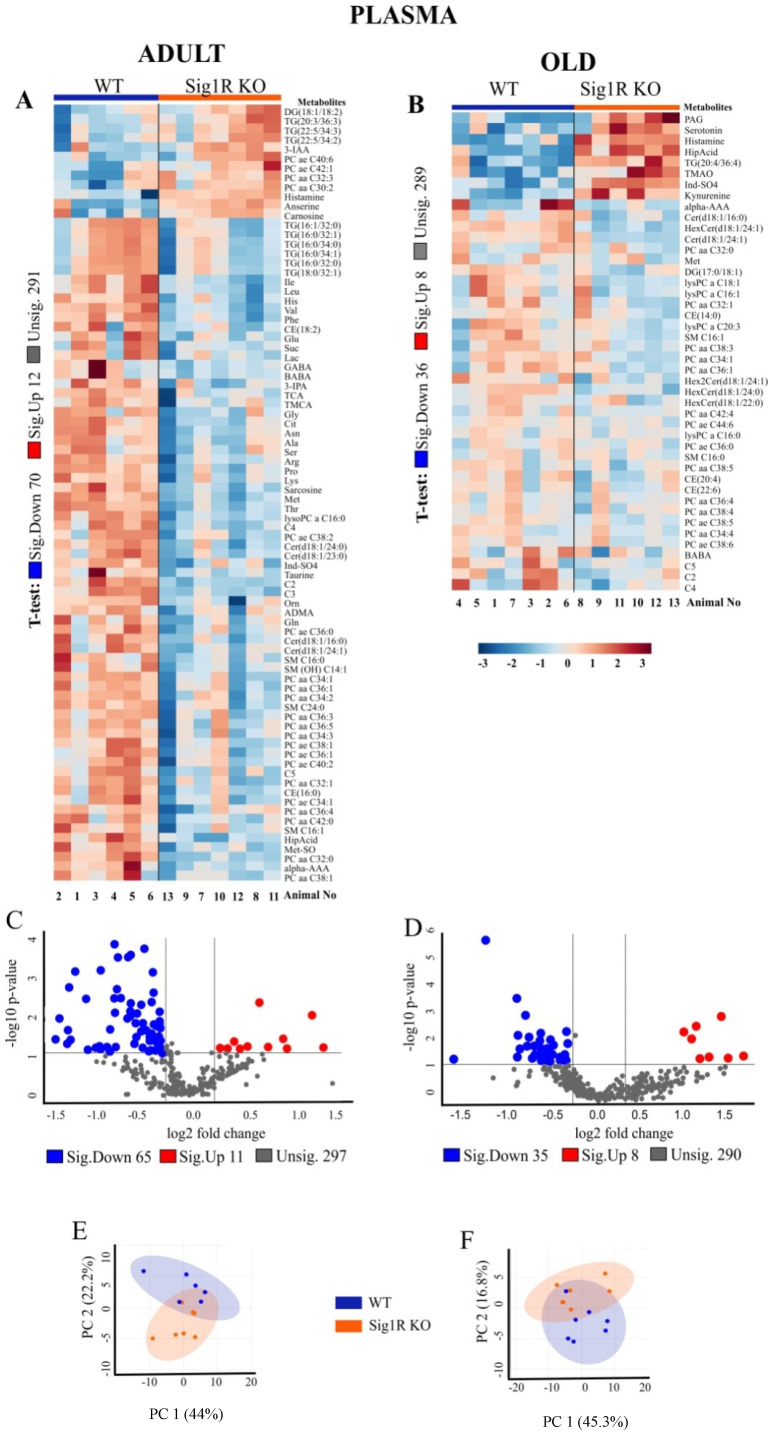
Metabolomic analysis revealed decreased amino acids and effects on related metabolite pathways in Sig1R KO mice. (**A**) Heatmap analysis highlighting metabolites showing the significant changes in plasma between adult WT and Sig1R KO mice. (**B**) Heatmap analysis highlighting me-tabolites showing the significant changes in plasma between old WT and Sig1R KO mice. (**C**,**D**) Volcano plot showing statistical significance (*p* value < 0.05) versus the magnitude of change (over 1.2) in plasma. (**E**,**F**) Score plots of PCA-discriminated metabolites in plasma between the WT and Sig1R KO groups. (**G**) Heatmap analysis highlighting metabolites showing the significant changes in brain tissue between adult WT and Sig1R KO mice. (**H**) Heatmap analysis highlighting metabolites showing the significant changes in brain tissue between old WT and Sig1R KO mice. (**I**,**J**) Volcano plot showing statistical significance (*p* value < 0.05) versus the magnitude of change (over 1.2) in brain tissue. (**K**,**L**) Score plots of PCA-discriminated metabolites in brain tissue between the WT and Sig1R KO groups. (**M**) The metabolome view shows all matched pathways according to the *p* values from the pathway enrichment analysis and pathway impact values from the pathway topology analysis. (**N**) Summary plot for quantitative enrichment analysis (QEA). Plasma and brain tissue samples were collected from 6- and 18-month-old WT and Sig1R KO mice (*n* = 6–7 mice/group). Analyses were produced using MetaboAnalyst 5.0. Sig.—significantly, Unsig.—unsignificantly.

**Figure 3 biomedicines-10-01572-f003:**
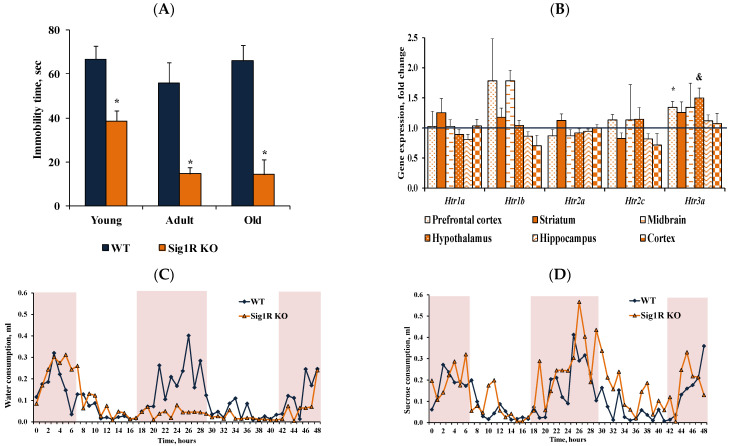
Reduced behavioral despair was associated with increased serotonin concentrations in Sig1R KO mice. (**A**) Immobility time (s) of young (2 months), adult (6 months), and old (15 months) WT and Sig1R KO mice was measured using the tail suspension test (*n* = 6–15 mice/group; two-way ANOVA followed by the Sidak test). (**B**) 5-HT receptor gene (*Htr1a, Htr1b, Htr2a, Htr2c,* and *Htr3a*) expression in the prefrontal cortex, striatum, midbrain, hypothalamus, hippocampus, and cortex of adult (4 months) WT and Sig1R KO mice (*n* = 8 mice/group; unpaired t test). (**C**) Water consumption of old (14 months) Sig1R KO (*n* = 6 mice/group) and WT (*n* = 8 mice/group) male mice was measured using the PhenoMaster system (shaded area indicates dark hours 8 a.m.–8 p.m.). (**D**) Sucrose solution consumption of old (14 months) Sig1R KO (*n* = 6 mice/group) and WT (*n* = 8 mice/group) male mice was measured using the PhenoMaster system (shaded area indicates dark hours, 08:00–20:00). (**E**) Combined consumption of water and sucrose solution (% of body weight) by young (2 months) and old (14 months) mice was measured using the PhenoMaster system (*n* = 6–13 mice/group; unpaired t test). (**F**) Calculated sucrose preference (%) of young (2 months) and old (14 months) mice was measured using the sucrose preference test (*n* = 6–13 mice/group; unpaired t test). (**G**) Serotonin concentration in the plasma of adult (6 months) and old (18 months) Sig1R KO and WT mice (*n* = 6–7 mice/group; unpaired *t* test). (**H**) Serotonin concentration in brain tissue of adult (6 months) and old (18 months) Sig1R KO and WT mice (*n* = 6–7 mice/group; unpaired *t* test). (**I**–**K**) Anatomical position for 5-HT and (**L**–**N**) TPH2 staining was validated using the Allen Mouse Brain atlas (http://mouse.brain-map.org/static/atlas, accessed on 18 May 2020). (**J**) Measured optical density (OD) of the staining intensity of 5-HT in the cortex (*n* = 6 mice/group; two-way ANOVA followed by Tukey’s test). (**K**) Representative histological images of 5-HT staining in the cortex (scale bar = 300 µm). (**M**) Number of TPH2-stained cells in the dorsal raphe nucleus (DRN) (*n* = 6 mice/group). (**N**) Representative histological images of TPH2 staining in the DRN (scale bar = 100 µm). Data are shown as the mean ± SEM (* *p* < 0.05 vs. WT; & *p* < 0.06 vs. WT; # *p* < 0.05 vs. same water consumption group).

**Figure 4 biomedicines-10-01572-f004:**
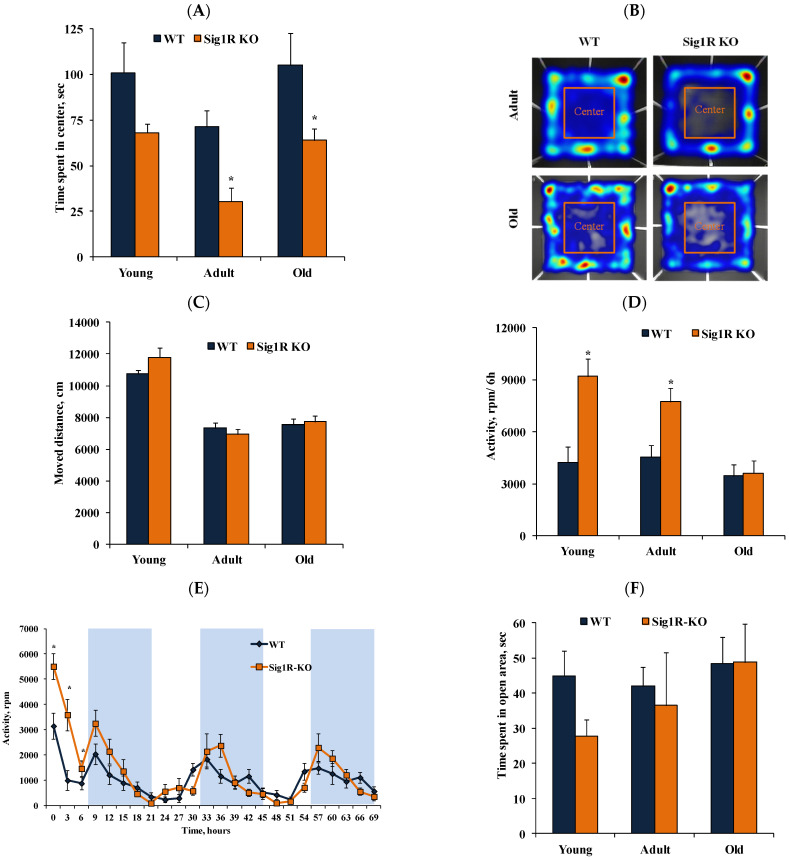
Anxiety-like behavior in Sig1R KO mice. (**A**) Time spent in the center of the arena of young (2 months), adult (5 months), and old (13 months) WT and Sig1R KO mice was measured using the open-field test (*n* = 6–10 mice/group; two-way ANOVA followed by the Sidak test). (**B**) Representative image of moved paths during the open-field test. (**C**) The movement distances of young (2 months), adult (5 months), and old (13 months) WT and Sig1R KO mice were measured using the open-field test (*n* = 6–10 mice/group). (**D**) Voluntary activity in the running wheel during the first 6 h of the 72-h trial of young (2 months), adult (4 months), and old (14 months) WT and Sig1R KO mice (*n* = 6–9 mice/group; two-way ANOVA followed by the Sidak test). (**E**) Representative curve of 72 h voluntary activity of young (2 months) mice in the running wheel (shaded area indicates dark hours, 08:00–20:00) of metabolic cages. (**F**) Time spent in the open area by young (2 months), adult (6 months), and old (17 months) WT and Sig1R KO mice was measured using the zero-maze test (*n* = 6–15 mice/group). Data are shown as the mean ± SEM (* *p* < 0.05 vs. WT).

**Figure 5 biomedicines-10-01572-f005:**
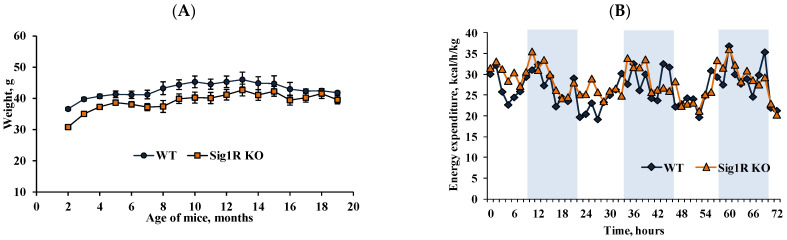
Weight and energy expenditure in Sig1R KO and WT mice. (**A**) Body weight gain throughout their lifetime. (**B**) The energy expenditure profile was measured within 72 h for 2-month-old mice (shaded area indicates dark hours, 08:00–20:00). Data are shown as the mean ± SEM (*n* = 5–11 mice/group).

**Table 1 biomedicines-10-01572-t001:** Gene expression in adult Sig1R KO mice brain.

Gene Name	*Sigmar1*	*Pgrmc1*	*Tph2*	*Gabbr2*	*Drd1*	*Drd2*	*Dnajc14 (Drip78)*
**Protein Name**	**Sigma-1 Receptor**	**Sigma-2 Receptor**	**Tryptophan 5-Hydroxylase 2**	**GABA-B Receptor Subunit 2**	**D(1A) Dopamine Receptor**	**D(2) Dopamine Receptor**	**Dopamine Receptor Interacting Protein**
**Prefrontal cortex**	-	1 ± 0.0	1.1 ± 0.1	1.1 ± 0.1	0.9 ± 0.2	0.7 ± 0.3	0.7 ± 0.2
**Striatum**	-	0.9 ± 0.1	1.5 ± 0.2	1.6 ± 0.3	1 ± 0.1	1 ± 0.2	1.1 ± 0.5
**Hypothalamus**	-	1 ± 0.1	0.9 ± 0.2	1 ± 0.1	1.1 ± 0.2	1 ± 0.1	0.6 ± 0.1
**Midbrain**	-	1.1 ± 0.2	0.8 ± 0.2	0.8 ± 0.1	1.8 ± 0.6	1 ± 0.1	1.1 ± 0.2
**Hippocampus**	-	0.8 ± 0.1	1.3 ± 0.2	0.9 ± 0.1	0.9 ± 0.1	0.9 ± 0.1	0.8 ± 0.0
**Cortex**	-	0.8 ± 0.3	1 ± 0.1	1.5 ± 0.2 *	0.7 ± 0.2	0.7 ± 0.3	1 ± 0.2

The relative expression levels for each gene were calculated with the ΔΔ*C_t_* method, normalized to the expression level of *β-actin*, and compared to the expression levels in WT control mice. Data are expressed as the mean fold change ± SEM (*n* = 8 mice/group). * *p* < 0.05 vs. WT mice; unpaired *t*-test.

**Table 2 biomedicines-10-01572-t002:** Organ weight versus body weight coefficients.

Organs/Group		Organs Weights, % of Body Weight		
		4 M	6 M	18 M
**Brain**	WT	1.2 ± 0.02	1.2 ± 0.05	1.3 ± 0.02
	Sig1R KO	1.3 ± 0.03 *	1.3 ± 0.08 *	1.4 ± 0.03 *
**Heart**	WT	0.5 ± 0.01	0.5 ± 0.03	0.5 ± 0.02
	Sig1R KO	0.4 ± 0.01	0.4 ± 0.01 *	0.4 ± 0.05 *
**Kidney**	WT	1.2 ± 0.05	1.6 ± 0.07	1.6 ± 0.06
	Sig1R KO	1.2 ± 0.03	1.3 ± 0.03 *	1.3 ± 0.05 *
**Pancreas**	WT	0.7 ± 0.07	1.0 ± 0.11	0.9 ± 0.07
	Sig1R KO	0.8 ± 0.06	0.5 ± 0.08 *	0.6 ± 0.05 *
**Liver**	WT	4.5 ± 0.09	5.2 ± 0.43	5.2 ± 0.33
	Sig1R KO	4.4 ± 0.10	4.8 ± 0.35	4.9 ± 0.35

Data are shown as the mean ± SEM (*n* = 5–10 mice/group). * *p* < 0.05 vs. WT mice.

## Data Availability

Data are available from the authors under reasonable request.

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
