# Peer review of "Antidepressive-like Behavior-Related Metabolomic Signatures of Sigma-1 Receptor Knockout Mice"

_biomedicines, 2022, doi:10.3390/biomedicines10071572_

Round 1

Reviewer 1 Report

It is a very interesting study in which the authors present metabolomic profiles of adult and old sigma-1 receptor (Sig1R) knockout mice and the age matched wild type mice. The authors report the effects of Sig1R knockout on mood-related behavior (depression and anxiety). The findings of this study suggest that Sig1R affects behavior by altering serotonergic and histaminerginc systems and the sphingolipid metabolic pathway.

Introduction provides sufficient background for the study. Methods are sound, appropriate and clearly described. Results are very well presented. Discussion is sound and comprehensive.

Author Response

Thank you very much for the effort and kind words

Reviewer 2 Report

The authors examined the metabolic profile in the Sig1R knockout (KO) mice in adulthood and old age. These findings were correlated with the assessment of depression- and anxiety-related behaviors. Overall, the manuscript is well written but some concerns should be addressed to improve its scientific soundness. 

Title: A title more exciting could increase the interest of a broader audience. For example: “ Sigma-1 Receptor knockout mice induce antidepressant-like actions related with specific metabolic changes”.

Material and methods: Authors should clarify what experiments were performed with anaesthetized animals with Ketamine. Antidepressant-like effects of Ketamine are known and could affect to the results observed. Please clarify if Ketamine was used for both, the detection of metabolites and Immunohistochemistry? 

Results:

- Name 5-HT receptors as 5-HT1A, 5-HT2A... (line 350). Cite of Sharp and Barnes is missing (line 353).  

- Authors performed two depression-related tasks. In this way, authors could indicate antidepressant-like properties (involving despair and anhedonia as reported). However, with only the results of the open field the authors indicate antigenic-like properties. No effects were observed in Zero maze test, and no Elevated plus maze for example was performed, more specific for anxiety detection. Although distance moved in the OF was no modified, the activity in the running wheel was increased. This could indicate an influence of locomotor activity on the OF. Thus, authors might explain better in the discussion this findings and no link directly to an “antigenic-like” behaviour. 

- In Figure 4: A and F look the same graphs. The graph for the Zero-maze test seems missed since authors indicate that there was no differences.

- Did Authors perform immunohistochemistry on the hippocampus or amygdala? They are relevant brain regions for depression and anxiety responses.  

Author Response

We are pleased that the Reviewer expressed interest in the subject and our approach. We have carefully followed through with the comments and suggestions of the Reviewer and corrected our manuscript accordingly. We hope that the manuscript could now be reconsidered for publication in Biomedicines.

Round 2

Reviewer 2 Report

Now the authors addressed all my concerns and, In my opinion, the manuscript may be published in this form.